# Psychosocial and Social Environmental Factors as Moderators in the Relation between the Objective Environment and Older Adults’ Active Transport

**DOI:** 10.3390/ijerph18052647

**Published:** 2021-03-05

**Authors:** Linda M. Nguyen, Lieze Mertens

**Affiliations:** 1Faculty of Health, Medicine & Life Sciences, Maastricht University, Universiteitssingel 40, 6229 ER Maastricht, The Netherlands; lindanguyen395@gmail.com; 2Department of Movement and Sport Sciences, Faculty of Medicine and Health Sciences, Ghent University, Watersportlaan 2, B-9000 Ghent, Belgium; 3Research Foundation—Flanders (FWO), Egmontstraat 5, B-1000 Brussels, Belgium

**Keywords:** walking, cycling, elderly, neighborhood, interaction effects, socio-ecological model

## Abstract

In order to develop tailored interventions aiming to encourage active transport among older adults, it is important to gain insights into the modifiable moderators affecting active transport behavior considering the neighborhood in which one lives. Therefore, this study aimed to determine which objective physical environmental factors have an impact on the active transport behavior of Belgian older adults (≥65 years old) and which psychosocial and social environmental moderators influence those relationships. Data from 503 independent living older adults who participated the Belgian Environmental Physical Activity Study in Seniors were included. Multilevel negative binominal regression models (participants nested in neighborhoods) with log link function were fitted for the analyses. Our resulted indicated that older adults living in an environment with higher residential density, higher park density, lower public transport density, and more entropy index had higher active transport levels. Furthermore, different types of neighborhood in which older adults live can lead to different moderators that are decisive for increasing older adults’ active transport behavior. Therefore, based on our results some recommendations towards tailored interventions could be given to increase older adults’ active transport behavior depending on the environment in which one lives.

## 1. Introduction

Despite the well-known physical, mental and social health benefits of regular physical activity (PA), 62% of the older adults living in Flanders (Belgium) aged from 65- to 74-year-olds and 87% of those aged 75 years [1] do not meet the recommended 150 min of moderate-to-vigorous physical intensity activity (MVPA) per week [1,2,3]. Walking and cycling are considered as accessible and inexpensive forms of MVPA [4,5] and are well-liked by Flemish older adults [6]. It was shown that regular walking and cycling (equivalent to 150 min of MVPA/week) were related to a decreased risk of all-mortality of 11% and 10% respectively, after adjustment for other PA [7]. As older adults eventually have to stop driving their car due to physiological restrictions (e.g., hearing, reaction time, muscle strength, vision), acceptable transportation alternatives to driving must be sought, of which active transport (AT) might be one [8,9]. AT or walking and cycling to a destination (e.g., to do groceries, to visit a friend) can be considered as an activity that is easy to integrate into older adults’ daily routines [10,11], benefit one’s health condition, and it is also beneficial for the economics and environment in terms of reduction in noise, traffic jams, and CO_2_-emissions [5]. However, in Belgium, respectively hardly 17% and 9% of the older adults (≥65 years) indicated walking or cycling as their main means of transport. Knowing that 22%, 47% or 63% of the trips older adults make are shorter than respectively 1, 3 and 5 km [12] AT is still not sufficiently embodied yet into their daily life routines.

AT can be influenced by both individual and environmental factors [13]. Individual factors include influences within the individual, such as demographics, biological, or psychological factors (i.e., self-efficacy, perceived benefits or barriers) and/or that are close to the individual, such as the social climate at home or in a neighborhood (i.e., social environmental factors) [13,14]. Environmental factors include influences that are beyond the individual, such as the presence of pedestrian/cycling facilities, parking, and traffic (i.e., physical environmental factors) [13,14]. AT can thus be explained by an interplay between several types of factors at multiple levels: individual (i.e., socio-demographic and psychosocial factors), social environmental, and physical environmental factors [13].

In a recent review and meta-analysis, strong evidence was found between the neighborhood physical environment and older adults’ walking for transport [15]. Whilst, limited evidence was available on the combination of walking and cycling for transport or cycling for transport solely [15]. Moreover, neighborhood environmental factors were more frequently assessed using self-reports instead of objective measures [15,16,17]. However, it is important to gain insights into determinants affecting AT behavior considering the neighborhood in which one lives. As the objective physical environment is often difficult to alter in short term, it is important to determine which individual and social environmental factors can be modified that may increase older adults’ AT behavior. In that way tailored interventions can be developed depending on the environment in which one lives. Unfortunately, knowledge about modifiable moderators between environment-AT associations is still lacking [15]. We hypothesized that older adults living in a less encouraging objective determined neighborhood will engage more in AT if they have a positive psychosocial view towards PA (i.e., more self-efficacy, perceiving more benefits/less barriers towards PA) [18], or live in a supportive social physical environment (i.e., a trustworthy and socially cohesive neighborhood) [19,20]. It is known from previous studies that self-efficacy, perceptions about benefits and barriers towards PA were observed as the major determinants of PA among older adults [21,22]. Furthermore, research on the social environmental factors of PA, especially among older adults, is still limited. Hence, more studies about determinants and moderators concerning environment-AT associations are needed to gain complete insights.

Therefore, the aim of this study was to determine which objective physical environmental factors have an impact on the AT behavior of older adults (≥65 years old) and which psychosocial and social environmental moderators influence those relationships.

## 2. Materials and Methods

### 2.1. Study Design

The cross-sectional data derived from the Belgian Environmental Physical Activity Study in Seniors (BEPAS Seniors) [23] were used for the current study. The BEPAS Seniors was a cross-sectional study which focused on independent living older adults aged 65 and over, living across twenty different neighborhoods in Ghent and its suburbs (Flanders, Belgium). This study aimed to examine the relationship between the neighborhood in which older adults live and their PA levels. The study protocol was approved by the Ghent University Hospital Ethics Committee. The BEPAS Seniors is detailed described in the original study conducted by Van Holle and colleagues [23].

### 2.2. Sampling

In short, data collection was performed between October 2010 and September 2012. Neighborhoods were stratified on Geographical Information System (GIS) -based data in Ghent (Flanders, Belgium) to determine neighborhood walkability (high vs. low) and matched on neighborhood annual household income level (high vs. low) [23]. Four types of neighborhood strata resulted: high walkability-high income, high walkability-low income, low walkability-high income, and low walkability-low income. In total twenty different neighborhood samples in Ghent and its suburbs were selected; five neighborhoods were allocated under each type of neighborhood.

Subsequently, older adults were randomly recruited in each type of neighborhood stratified based on gender and age (<75 versus ≥75 years old). Selected older adults were approached by sending an information letter by postal mail in which the aim of the study was indicated including the announcement of the home visit by a trained interviewer during the next two weeks. During the home visits, and after signing the informed consent, a face-to-face interview collected information about socio-demographic, self-reported physical functioning, residential self-selection, PA levels in the last week, psychosocial factors, and perceived social environmental factors. The following inclusion criteria were applied: aged ≥65 years old, able to understand and speak Dutch/Flemish, able to walk a few hundred meters without heavy difficulties in physical functioning and lived independently.

### 2.3. Measures

#### 2.3.1. Socio-Demographic Factors and Residential Self-Selection

During the face-to-face interview, the following participants’ information was retrieved: age, gender, origin (yes: Belgium as country of birth vs. no), education (tertiary vs. no tertiary level), occupation (household, blue-, or white collar), living situation (with vs. without partner) and owning motorized vehicles (at least one vehicle vs. no). Participants’ self-reported physical functioning was derived from the Short Form 36 item Survey (SF-36) [24]. Participants had to denote on a 3-point scale (severely limited; somewhat limited; not limited) to what extent they were physically limited while performing ten general activities, such as lifting or carrying groceries, climbing stairs, and walking short distances. Subsequently, activities in which participants reported to be severely or somewhat limited were summed. Participants’ physical functioning, ranging from zero to ten, was obtained by reversing this latter variable. Residential self-selection was assessed based on an eight-item scale derived from a study [25]. Respondents were asked to rate how important eight possible reasons (e.g., proximity of open spaces, sense of community) were for selecting their neighborhood. Internal consistency of the scale in the current sample was good (Cronbach’s alpha = 0.83).

#### 2.3.2. Self-Reported Active Transport Behavior

The dependent variable, AT, was measured using the long version of the International Physical Activity Questionnaire (IPAQ) [26,27]. The frequency (i.e., number of days) and duration (i.e., average time/day) of walking and cycling for transport in the last seven days was assessed. The IPAQ was proven reliable and valid specific among older adults [27], whereby its test-retest reliability was overall moderate to good [28].

#### 2.3.3. Psychosocial Factors towards PA

In total six psychosocial factors towards PA were included in the current study. A more detailed description of the content of the items and scoring of the factors can be found in the original study of Van Holle and colleagues [29]. In short, self-efficacy, consisted of five items, referred to one’s confidence to be physically active in difficult circumstances. Perceived benefits, likewise, consisted of five items, reflected on benefits, such as health benefits and meeting new people. Perceived barriers, consisted of seven items, reflected on barriers, such as feeling for ‘not being good enough’ and bad weather conditions. Furthermore, social support indicated the partner’s/friends’ supportive attitude towards PA. Social norm indicated what the partner/friends think(s) about one should perform PA. Lastly, modeling referred to what extent the partner/friend(s) is/are being physically active. Therefore, these last three factors consisted of each two items. The items were based on existing validated questionnaires used among Belgian adults [30,31]. Those psychosocial factors were assessed on a 3-, 5- or 7-point scale, see Table 1 for a detailed overview. Each factor variable was calculated by averaging the scores on the items and considered as continuous variable.

#### 2.3.4. The Perceived Social Neighborhood Environment

In total four perceived social environmental factors were included in the current study: talking to neighbors, social interactions with neighbors, neighborhood social trust and cohesion, and neighborhood social diversity. ‘Talking to neighbors’ referred to informal social interactions with neighbors and consisted of two items, while ‘social interaction with neighbors’ referred to formal social interactions and consisted of three items. These two factors measured the extent of social interactions between older adults and their neighbors [32,33]. ‘Neighborhood social trust and cohesion’ was derived from a questionnaire by Sampson [34] and consisted of four items assessed participants’ agreement about their local neighborhood (e.g., ‘people in this neighborhood can be trusted’). ‘Neighborhood social diversity’ gives an indication of the social composition of the neighborhood (e.g., proportion of immigrants, youngsters, older people in the neighborhood), consisted of three items and was assessed analogous to previous research in older adults [35]. Each factor variable was calculated by averaging the scores on the items and considered as continuous variable. See Table 1 for a detailed overview concerning the scoring of the factors. A more detailed description of the content of the items can be found once again in the original study of Van Holle and colleagues [36].

#### 2.3.5. The Objective Physical Neighborhood Environment

The BEPAS Seniors included objective GIS-data for all participants using a sausage buffer of 500 m around their home address [23]. The near and direct neighborhood environment may be important for older adults, as they are more likely to walk/cycle in streets they are familiar with and they are overall less commutable to other locations compared to other age groups [15,21,37,38]. The density of the following five physical environmental factors were objectively calculated in a 500 m buffer zone surrounding each participant’s home address: residential density, park density, public transport density, intersection density and entropy. First, residential density was described as the ratio between the number of single- and multifamily domiciles and the area of all parcels within or partial within the buffer zone. Second, park density was defined as the ratio between the number of parks that are full or partial in the buffer zone and the total land area in the buffer zone. Third, public transport density was described as the ratio between the number of transport stops (i.e., bus, tram, train) and the total buffer zone. Additionally, intersection density was the ratio between the number of three- or more-way crossings and the total buffer zone. Lastly, entropy referred to the index of land use mix diversity within the buffer zone [39]. The entropy index was calculated by using Dobesova’s and Krivka’s formula [39]:$$H\left(S\right)=\;\frac{-\sum_{i=1}^{k}\left[\left(\rho_{i}\right)\cdot \left(\ln\rho_{i}\right)\right]}{\ln k}$$
where *H*(*S*) indicates the entropy index (also called Shannon index), *ρ_i_* is the area of a category of land use over the total area of all categories (within one neighborhood), and *k* is the number of land use categories in the particular neighborhood. Five land use categories were included in the formula: residential, commercial, civic, entertainment, food and private/public recreational.

### 2.4. Data Analyses

Descriptive statistics of the total study sample were obtained using SPSS 26.0 software (IBM Corp, Armonk, New York, United States). Generalized estimating equations (GEE) in the SPSS software was used to examine the main associations between objective physical environmental factors and AT, and to determine the moderating effects of psychosocial and social environmental factors. Multilevel negative binominal regression models (participants nested in neighborhoods) with log link function were fitted for the analyses as the dependent variable, AT (combined minutes walking and cycling for transport per week), was positively skewed and contained a considerable number of null values (31.4%).

First, single predictor models (i.e., two-level negative binominal regression models) for each potential covariate (i.e., age, gender, living status, education, vehicles, physical functioning and residential self-selection) were fitted. Only the significant covariates (i.e., living status and physical functioning), see Table A1 in Appendix A, were added as covariates in the further analyses, given the complexity of the models. Second, single predictor models for each objective environmental factor, for each psychosocial and social environmental factor were separately fitted and added in Appendix A (see Table A1) for completeness. Third, 50 single interactions models were separately estimated between the objective physical environmental factors and each potential moderator (i.e., psychosocial and social environmental factors), and can be found in Appendix A
Table A2. Fourth, all single interaction effects from the third step surpassing the statistical threshold of *p* < 0.10 were simultaneously added in the final model. The final model is presented in Table 2. Only the significant interaction effects observed in the final model are further described in the text and visualized with graphs. To facilitate model convergence and interpretation, all objective environmental factors were standardized. Level of significance was defined at α = 0.10 (trend) and α = 0.05.

## 3. Results

### 3.1. Descriptive Statistics

In total, 503 older adults ranged in age from 65 to 97 years old participated the BEPAS Seniors study. The socio-demographics, the self-reported AT behavior and the psychosocial, social environmental, and objective physical environmental factors of the total study sample are presented in Table 1. Just over half of the sample (53.4%) were women, 38.0% of the participants performed tertiary education, and 65.3% lived with a partner. On average, the sample performed 121.8 ± 163.6 min per week AT, and 31.4% of the total study sample did not engage in AT.

### 3.2. Final Model

Main Effects Objective Environmental Factors

The final model is presented in Table 2. Older adults living in an environment with higher residential density, higher park density, lower public transport density, and more entropy (trend) had significantly higher AT levels. Living in a neighborhood that is one standard deviation higher in residential density was associated with 263% more minutes AT per week. Older adults living in a neighborhood that is one standard deviation higher in park density were associated with 35% more AT per week. Older adults living in a neighborhood that is one standard deviation lower in public transport stops were associated with 69% more AT. Living in an environment with one standard deviation higher in entropy was marginally associated with 389% more AT.

### 3.3. Interactions with Psychosocial, and Social Environmental Factors

The association between objective residential density and AT was marginally significantly moderated by self-efficacy (*p* = 0.067). Older adults living in low residential density neighborhoods with high self-efficacy towards PA performed more AT in comparison to older adults perceiving low self-efficacy towards PA. In high residential neighborhoods, older adults with low self-efficacy performed more AT in comparison to older adults perceiving high self-efficacy (see Figure 1).

The association between objective park density and AT was marginally significantly moderated by perceived barriers towards PA (*p* = 0.088). The positive effect of park density on AT was greater for older adults who perceived low barriers towards PA in comparison to older adults perceiving high barriers towards PA (see Figure 2).

The association between objective public transport density and AT was significantly moderated by perceived benefits towards PA (*p* = 0.002). Older adults living in low transport dense environments with low benefits towards PA performed more AT in comparison to older adults with high benefits towards PA. While, in high dense environments, older adults with high benefits towards PA performed more AT in comparison to older adults with low benefits towards PA (see Figure 3).

The association between objective public transport density and AT was marginally significantly moderated by talking to neighbors (*p* = 0.080). The negative effect of transport dense environments on AT is greater for people who talked less with their neighbors in comparison to older adults who talked more with their neighbors (see Figure 4).

The association between objective entropy and AT was significantly moderated by barriers towards PA (*p* = 0.046). Only for older adults with high barriers there was a difference depending on the entropy index of the neighborhood on AT. Older adults living in low entropy index neighborhoods with high barriers towards PA performed less AT in comparison to older adults with low barriers towards PA. While older adults living in high entropy index neighborhoods with high barriers towards PA performed more AT in comparison to older adults with low barriers towards PA (see Figure 5).

The association between objective entropy and AT was significantly moderated by benefits towards PA (*p* = 0.015). Only for older adults with low benefits towards PA there was a difference depending on the entropy index of the neighborhood on AT. Older adults living in low entropy index neighborhood with low benefits towards PA performed less AT in comparison to older adults with high benefits towards PA. While older adults living in high entropy index neighborhood with low benefits towards PA did more AT in comparison to older adults with high benefits towards PA (see Figure 6).

## 4. Discussion

The aim of this study was to determine which objective physical environmental factors have an impact on the AT behavior of older adults (≥65 years old) and which psychosocial and social environmental moderators have an impact on those relationships. Our results indicated that older adults living in an environment with higher residential density, higher park density, lower public transport density, and more entropy had higher AT levels. Similar results among older adults were found in a recent meta-analysis but for the perceived measure of residential density, park availability and land use mix diversity [15].

Residential density and land use mix diversity are two components of walkability, which reflects the ease of walking to destinations in a neighborhood [40]. Among Belgian older adults, high neighborhood walkability was found to be related to higher levels of transportation walking experiences [23]. However, for the third component of walkability, intersection density or street connectivity, we did not find a significant association with AT, while in the literature objective [15] or perceived [22] street connectivity was often correlated to more AT. Van Cauwenberg [22] indicated that a higher street connectivity could make AT more attractive as older adults are supported to choose more alternative routes (e.g., avoid busy streets or steep slopes) [22]. A possible explanation why we did not find this association in our study might be the small buffer size (i.e., 500 m). Even though using a small buffer size for older adults is recommended [15,37,38], it might be interesting to look at different or larger buffer sizes. Furthermore, contrary to our expectations and previous results, a contradictory result was found for public transport accessibility. Our results indicated that low public transport dense neighborhoods were associated with more AT, while a positive association was found previously [15]. A possible explanation for this finding might be that older adults have to walk further distances to access destinations because they do not have access to public transport. This might be a sign that those neighborhoods need more investment in public transport accessibility. Given the conflicting results with the literature, these findings need further exploration.

In order to better respond to such relationships with the purpose of interventions, it is good to have insight into the possible moderators between environment-AT associations. We hypothesized that older adults living in a less encouraging objective determined neighborhood will do more AT if they have a positive psychosocial view towards PA [18], or live in a supportive social physical environment [19,20]. A first interesting result is that the negative effect found on AT according to high public transport dense neighborhoods could be reduced if older adults perceiving high benefits towards PA or when they often talk to their neighbors. According to the Health Belief Model [41], older adults’ AT behavior change is influenced by both perceived benefits and barriers of change. Therefore, for older adults living in high public transport dense neighborhoods the benefits towards PA should be more emphasized. Additional, if older adults talk more to their neighbors, they get more familiar with their neighborhood and nearby surroundings, which might increase their willingness to do more AT. A recent longitudinal study of Josey and Moore [42] among adults indicated that a more favorable personal social network may be essential for bringing change into physical inactivity behavior. Research on the social environmental factors of PA among older adults is still limited, therefore, further exploration on these findings are certainly needed.

Second, as stated in our results, park density increases AT, and also from previous research we know that good accessibility of parks (e.g., low traffic on route, slope of terrain <5%) increases the likelihood that older adults will visit a park [43,44]. Furthermore, our results indicated that the positive effect of park density on AT among older adults could be strengthened if they perceived low barriers towards PA (e.g., lack of time, bad weather conditions, fear of falling) in comparison to older adults perceiving high barriers towards PA. From a recent longitudinal study among older adults, we learn that by decreasing the perceived barriers towards PA the engagement in cycling for transport over time can be increased [17]. Moreover, previous cross-sectional research showed that perceived barriers and benefits were the most important correlates in relation to overall MVPA [29,45]. In other words, when older adults perceiving high barriers and these are not reduced, it might discourage them even more their intention to be physically active, despite living in higher park dense neighborhoods.

Third, we found that older adults living in low residential dense neighborhoods perceiving a higher self-efficacy will do more AT than older adults perceiving less self-efficacy. Previous studies confirmed the association between higher self-efficacy towards PA (e.g., confidence to be physically active when the weather is bad, or when you have little time) and higher levels of PA [46], or specifically higher levels of walking for transport [17]. Hereby, self-efficacy, the expectation older adults have about their own ability to do more AT [47], may be important to increase PA levels among older adults. More specifically, according to our results, we can conclude that improving the self-efficacy towards PA may be more essential among older adults living in low residential dense neighborhoods.

Fourth, an unexpected result was found according to the moderating effect of perceived barriers and benefits towards PA for the association between entropy and AT. The entropy index represents how homogenous or heterogeneous the usage of a particular area is [39]; the higher is the diversity of the land use, the higher is the entropy index. Previous longitudinal [17], cross-sectional qualitative [21,48] and quantitative [49,50,51] research reported similar findings among older adults concerning the positive main association between entropy or land use mix diversity and walking for transport. We found that older adults living in an environment with more entropy will marginally have higher levels of AT which corresponds to the literature. However, for older adults perceiving low barriers towards PA or perceiving high benefits towards PA, no difference will be found according to the entropy index of the neighborhood in which older adults live. Nevertheless, for older adults perceiving high barriers towards PA or perceiving low benefits towards PA, there will be a difference in AT depending on the entropy index of the neighborhood in which one lives. Older adults living in low entropy index neighborhood with high barriers or low benefits towards PA will do less AT in comparison to older adults perceiving low barriers or high benefits towards PA. While older adults living in high entropy index neighborhood with high barriers or low benefits towards PA do more AT in comparison to older adults with low barriers or high benefits towards PA. In other words, older adults who consider AT as more beneficial to them and have lower barriers towards PA will do it regardless of where they live, while older adults who think AT is less beneficial and have more barriers, would not do it in an environment that is not conducive to it. Consequently, a more entropy dense neighborhood enables walking and makes it easy, even for older adults who would not normally walk or cycle. Therefore, older adults living in neighborhoods where the diversity of land use mix is low, it is important to encourage benefits towards PA and decrease the barriers towards PA.

In summary, interventions aiming to promote AT are necessary, since a small number of older adults perform AT. It is important to be aware that different types of neighborhood in which older adults live can lead to different moderators that are decisive for increasing older adults’ AT behavior. Based on our results, the following suggestions could be made for future intervention. Interventions focusing on increasing the benefits towards PA as well as talking to neighbors might help to increase AT behavior of older adults living in high public transport dense neighborhoods which is negative correlated to older adults’ AT. For older adults living in high park dense neighborhoods, decreasing barriers towards PA could increase older adults’ AT. Furthermore, interventions focusing on providing self-efficacy training to older adults to boost their confidence and ability to do more physical activities might rise older adults’ AT living in low residential dense neighborhoods. Lastly, declining barriers and increasing benefits towards PA will enhance older adults’ AT behavior if they live in an environment with a low entropy index. Although, this study gives a first indication of possible changeable psychosocial and social environmental moderators in the relation between the objective physical environment and older adults’ AT, longitudinal studies are crucial to gain better insights into which moderators could predict the relationship over time.

The current study has several strengths and limitations. A first strength is the investigated moderating effects of psychosocial and social environmental factors which are modifiable factors that can be embedded in interventions aiming to encourage AT taking into account the environment in which one lives. This has hardly been studied before among older adults. Another strength is that the physical environmental factors were objectively assessed, while previous research often used subjectively assessed data which might be more sensitive for biases. Furthermore, older adults’ characteristics and factors were largely assessed through a face-to-face interview by trained health professionals. Responses can be more accurately obtained during a guided survey instead of a self-administered survey, especially among older adults as they may experiences more cognitive difficulties when filling in a questionnaire by themselves [52]. A validity study showed that less over-reporting bias occurred when an interview-administered version was used for IPAQ instead of a self-administered version [53]. Lastly, the large sample size enabled to investigate both main and moderating effects.

Limitations of the current study mainly include its cross-sectional design, so no causal inferences of the findings could be made [54]. It is recommended to perform longitudinal and/or experimental studies, which enables to investigate changes in AT behavior over time and to identify causality. Furthermore, psychosocial factors were questioned in relation to PA in general and not AT in specific, which might have led to inaccurate findings [55]. Nevertheless, these psychosocial factors were based on existing validated questionnaires used among Belgian adults [30,31]. Another limitation was the self-report of older adults’ AT behavior. GPS devices gives us the opportunity to objectively assess the transport behavior, however there are also concerns such as losing GPS contact or misclassify the travel mode [56,57]. Therefore, it is recommended to combine these GPS devices with activity diaries or accelerometer in order to fulfill the missing information [58,59]. Furthermore, we are aware that we have not included all possible variables that have an impact on active transport, such as vehicle traffic or pleasantness of routes for walking and cycling. Therefore, we have to be careful when drawing policy conclusions. Lastly, as this research is conducted in Belgium, the generalization of these results to Western Europe may be possible due to the similar cycling culture should be approached with caution.

## 5. Conclusions

Interventions aiming to promote AT are necessary, since a small number of older adults perform AT. It is important to be aware that different types of neighborhood in which older adults live can lead to different moderators that are decisive for increasing older adults’ AT behavior. Our results indicated that older adults living in an environment with higher residential density, higher park density, lower public transport density, and more entropy had higher levels of AT. Furthermore, we can give a few recommendations towards interventions based on the results found for the modifiable moderators which will give the opportunity to develop tailored interventions depending on the environment in which the older adult lives. First, increasing the benefits towards PA as well as talking to neighbors might help to increase AT behavior of older adults living in a high public transport dense neighborhood which is negative correlated to older adults’ AT. Second, decreasing barriers towards PA, could increase AT for older adults living in high park dense neighborhoods. Third, interventions focusing on providing self-efficacy training to older adults to boost their confidence and ability to do more physical activities might rise older adults’ AT living in low residential dense neighborhoods. Lastly, interventions focusing on declining barriers and increasing benefits towards PA will enhance older adults’ AT behavior if they live in an environment with a low entropy index.

## Figures and Tables

**Figure 1 ijerph-18-02647-f001:**
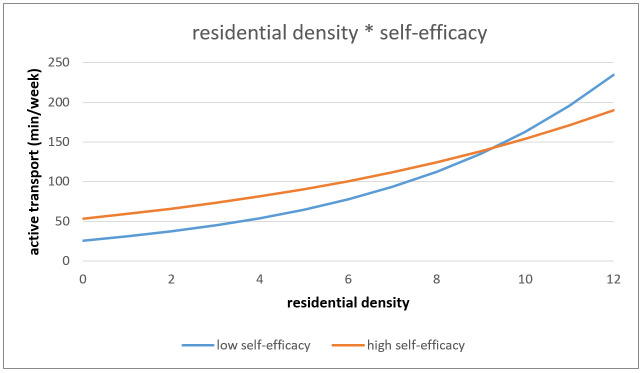
Interaction effect of residential density and self-efficacy on active transport.

**Figure 2 ijerph-18-02647-f002:**
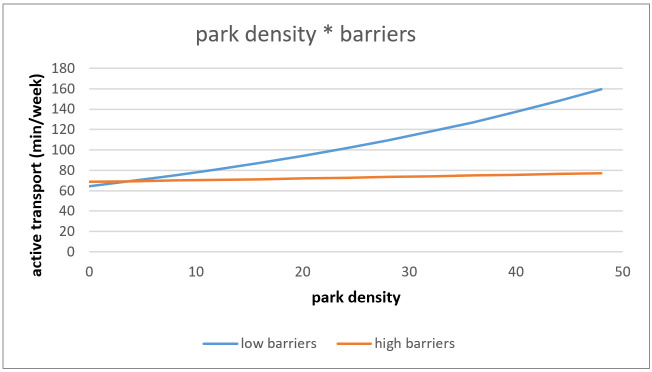
Interaction effect of park density and barriers towards PA on active transport.

**Figure 3 ijerph-18-02647-f003:**
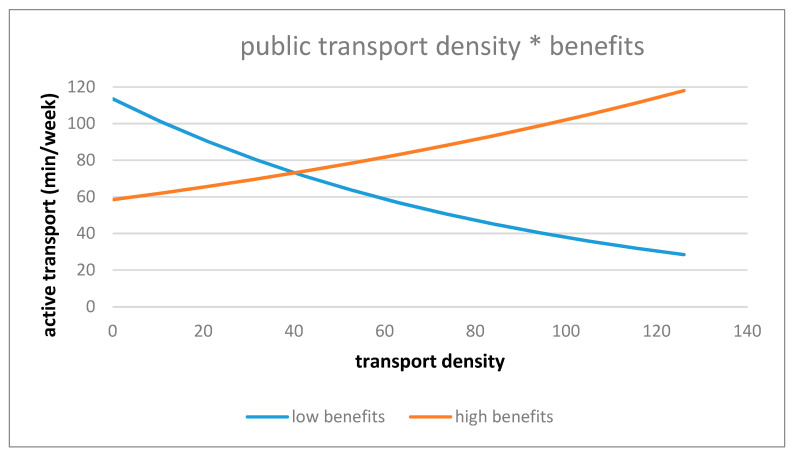
Interaction effect of public transport density and benefits towards PA on active transport.

**Figure 4 ijerph-18-02647-f004:**
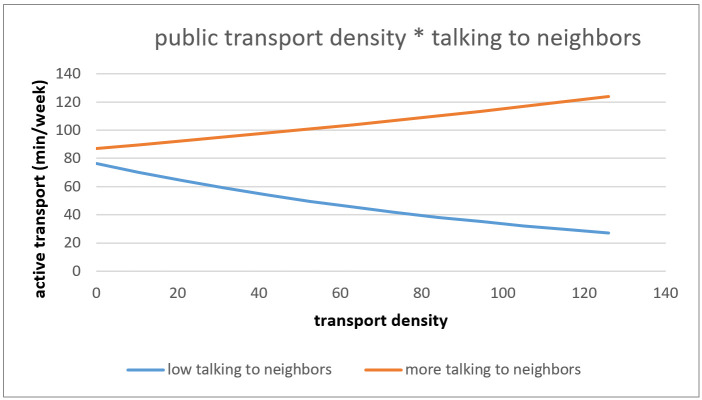
Interaction effect of public transport density and talking to neighbors on active transport.

**Figure 5 ijerph-18-02647-f005:**
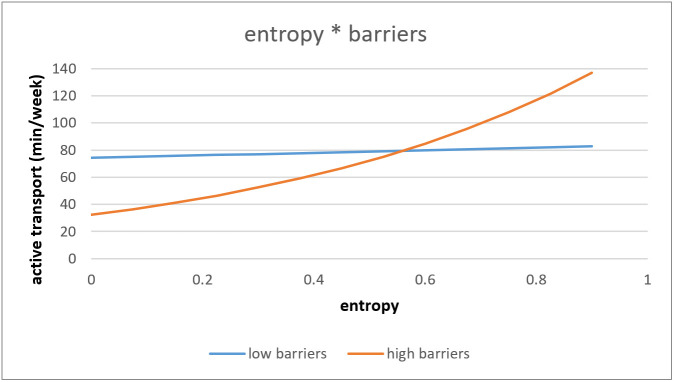
Interaction effect of entropy and barriers towards PA on active transport.

**Figure 6 ijerph-18-02647-f006:**
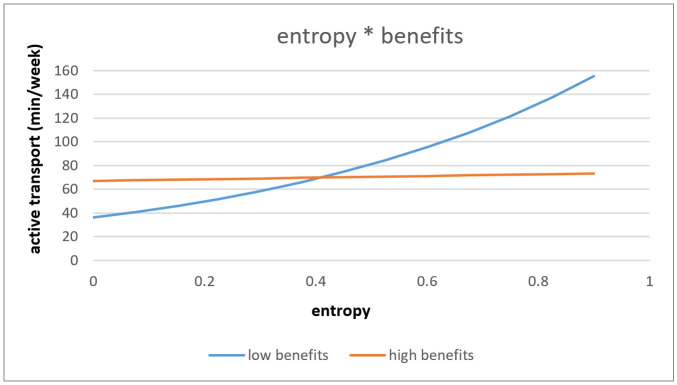
Interaction effect of entropy and benefits towards PA on active transport.

**Table 1 ijerph-18-02647-t001:** Descriptive statistics of the participants (*n* = 503).

Characteristics	Values
Age in years (M ± SD)	74.4 ± 6.2
Women (%)	53.4
Living with a partner (%)	65.3
Tertiary education (%)	38.0
Physical functioning (M ± SD)	7.05 ± 2.37
No motorized vehicles in the household (%)	20.8
Active transport (min/week) (M ± SD)	121.8 ± 163.6
No active transport (%)	31.4
**Psychosocial factors towards physical activity (M ± SD)**	
Self-efficacy ^a^	2.1 ± 0.5
Perceived benefits ^b^	3.4 ± 0.8
Perceived barriers ^b^	1.9 ± 0.7
Social norm ^b^	2.9 ± 1.4
Social support ^b^	2.8 ± 1.5
Modeling ^c^	2.8 ± 1.9
**Social neighborhood environmental factors (M ± SD)**	
Talking to neighbors ^c^	5.5 ± 1.4
Social interactions with neighbors ^c^	2.2 ± 1.1
Neighborhood social trust and cohesion ^d^	3.7 ± 0.8
Neighborhood social diversity ^b^	4.2 ± 0.7
**Objective neighborhood environmental factors, buffer 500 m (M ± SD)**	
Residential density ^e^	4871.1 ± 3261.1
Park density ^f^	10.9 ± 8.6
Public transport density ^g^	33.3 ± 19.7
Intersection density ^h^	162.7 ± 70.0
Entropy ^i^	0.5 ± 0.2

M = mean; SD = standard deviation; ^a^ = assessed on a 3-point scale; ^b^ = assessed on a 5-point scale; ^c^ = assessed on a 7-point scale; ^d^ = assessed on a 4-point scale; ^e^ = number of dwellings per surface buffer 500 m; ^f^ = number of public parks of all sizes per surface buffer 500 m; ^g^ = number of public transportation stops of any kind per surface buffer 500 m; ^h^ = number of intersections per surface buffer 500 m; ^i^ = range from 0 (= perfect homogenous land use) to 1 (= perfect heterogeneous land use).

**Table 2 ijerph-18-02647-t002:** Final model.

Factors	ExpB (95% CI)
Living status (ref: living with a partner)	0.97 (0.70–1.33)
Physical functioning	1.18 (1.14–1.23)
Residential density	2.63 (1.09–6.35) *
Park density	1.35 (1.02–1.77) *
Public transport density	0.31 (0.15–0.64) **
Intersection density	0.99 (0.72–1.36)
Entropy	3.89 (0.80–18.86) ^
Self-efficacy	1.40 (1.06–1.84) *
Perceived benefits	0.93 (0.84–1.04)
Perceived barriers	0.92 (0.76–1.11)
Social support	1.09 (1.00–1.19) *
Modeling	0.96 (0.90–1.03)
Neighborhood social trust and cohesion	1.15 (0.96–1.36)
Talking to neighbors	1.20 (1.04–1.38) *
Social interaction to neighbors	0.90 (0.81–1.00) *
Residential density * self-efficacy	0.78 (0.60–1.02) ^
Residential density * social support	1.02 (0.90–1.16)
Residential density * modeling	0.97 (0.91–1.03)
Residential density * social trust and cohesion	1.02 (0.81–1.02)
Park density * perceived barriers	0.90 (0.81–1.02) ^
Public transport density * perceived benefits	1.22 (1.07–1.38) **
Public transport density * talking to neighbors	1.08 (0.99–1.18) ^
Intersection density * social support	0.96 (0.87–1.06)
Entropy * perceived benefits	0.83 (0.71–0.96) *
Entropy * perceived barriers	1.25 (1.00–1.56) *
Entropy * neighborhood social trust and cohesion	0.80 (0.56–1.10)
Entropy * social interactions to neighbors	0.95 (0.87–1.03)

^ *p* < 0.10; * *p* < 0.05; ** *p* < 0.01; ref = reference category.

## Data Availability

The data presented in this study are available on request from the corresponding author.

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
