# Peer review of "Psychosocial and Social Environmental Factors as Moderators in the Relation between the Objective Environment and Older Adults’ Active Transport"

_ijerph, 2021, doi:10.3390/ijerph18052647_

Round 1
Reviewer 1 Report
1. Introduction is generally good. Language could be a little more clear. There are several typos and places where the English isn't quite right (throughout the paper). In much of the introduction you make broad claims that aren't necessarily true everywhere. In some cases its just a minor language change that is needed. For example, you state that older adults eventually have to give up driving (line 37) but in many places they do not and in the US driving agencies frequently renew licenses of people who perhaps shouldn't be driving but still do as active modes are not feasible. So just be sure to contextualize your comments like this throughout the introduction.
2. Line 46... 3 and 5 km seem somewhat far for active transport. Perhaps provide a broader context to situate the reader how this compares to the rest of the population. Looking at the citations, this appears to be Belgium specific, so again just be sure to mention that.
3. Final sentence in intro is key and makes the point of the paper very clear.
4. In methods (and abstract) you use the term 'community-dwelling' which is a little unclear to me what this means exactly. I'm guessing it means those living on their own NOT in assisted care type facilities. But perhaps specify or change terminology to something more universally recognized.
5. All of the data you used in this was collected through an earlier study, correct? Methodology in some places makes it unsure if you were the ones doing face-to-face interviews after utilizing the previous data set or if all that was part of original study. I'm assuming the latter, but again, be more clear what data was collected by you and what wasn't. If none of it was collected by you, this section does not need to be so long.
6. It is a little unclear in your methods what your units are for your different variables. You do a good job describing what each variable is, but from the descriptions and then the values in table 1, it is unclear what a 2.1 +/- 0.5 for self-efficacy means exactly, for example. And with density, is this value (4871) people per 500m area or dwelling units? Parks - is that number of parks or Hectares of parks? etc, etc. Table 1 could perhaps use a third column to ensure we know what exactly is being measured in each value for each variable.
7. The language of the results section makes following the results very difficult. There are a lot of variables to be keeping track of and you keep switching what you are taking about. and the way it is written makes the results sound more complicated than they actually are.
8. Table 2 should probably be two separate tables. First one would be the first half showing all the variables on their own to AT. And I'm not sure why you would bother reporting the first step of the psychosocial and social environment factors directly to AT since these are the moderating factors. I think perhaps a more useful way to present the objective factors*psychosocial/social environment in a matrix instead of a list. Basically your Objective variables on the left side of a table and then your psychosocial varaiales along the top. From the way it is presented now it is completely unclear why some values are reported and others are not...
9. Discussion is good. One thing I would perhaps expand on is why you think the low public transit density led to more AT and what this means. I would think it is because they are walking farther distances to access things simply because they do not have access to transit. This isn't a necessarily good thing from an accessibility standpoint since some people who may not be as able to walk those farther distances are still doing it because they have no alternative. This could be a sign that these areas need more investment in public transit. While your focus is on AT, it is important to remember that transport is usually a means to an end and that while we want to encourage and enable AT, we don't want to do so at the expense of putting undue transport burden on a segment of the population.
10. On your 4th point (line 364), although perhaps unexpected for you, this is not really that unexpected. Older adults (and adults in general) who see AT as more beneficial to them and have lower barriers will do it regardless of where they live. On the other hand, for those who think AT is less beneficial and have barriers, they won't do it in an environment that isn't conducive to it. Therefore, in a more dense environment that enables walking and makes it easy, even those who wouldn't normally walk do get AT. This I think is one of your most significant findings.
11. Overall, good paper. The start of it is a little clunky and takes a while to get to the meat of the paper - the findings and discussion. The discussion should be expanded and perhaps the methods section reduced. Some of the points made in the discussion are very good insights into the levels of AT that older adults will engage in. The last paragraph in your discussion makes it sound like you ran out of space to do these things, so perhaps shortening the methods section would be good. And not much would really be lost since the data collected was from a previous study.
Reviewer 2 Report
The authors mention that ‘the aim of this study is to determine which objective physical environmental factors have an impact on the AT behavior of older adults and which psychosocial and social environmental moderators influence those relationships.’ (Line 79-81) There are many notable studies exploring the relationships between objective physical environment and physical activity, as well as the relationships between the perceptions of objective physical environment and physical activity in various types of residential neighborhood. The contribution of this study, to my view, is to supplement the existing studies by adding both ‘psychosocial and social environmental factors as moderators in the relation between the objective environment and older adults’ active transport.’ (line 1-3). The aim is meaningful and the hypothesis (line 69-71) is intriguing. However, there are two major questions in the research design, which need to be clarified seriously.
- As revealed in the Table I (single-predictor models to AT) of Appendix A, when all factors are directly put as independent variables, many of demographic and objective physical factors’ p-values are significant, while most of psychosocial and social factors’ p-values are not (except ‘ talking to neighbors’ factor). This confirms demographic and objective physical factors have relatively more influences. Nevertheless, this does not mean psychosocial and social factors do not operate as independent variables. Considering psychosocial and social factors’ relatively lesser influences, it is meaningful and agreeable to position them as ‘moderators’ in this paper. However, as seen in the Table 2 (single interactions models to AT), when objective physical factors and the psychosocial as well as social factors are combined by rote, there are many issues. Among others, the multicollinearity between each facto and moderator needs to be carefully resolved. The existing psychosocial factors and physical* psychosocial factors as well as social factors and physical*social factors are already imbedded and operate in this model, which result in distortion. This needs to be clarified.
- This study mentions four types of neighborhoods in the data sampling, yet there are not enough explanation about how those types were defined. More detailed explanation and analysis about the neighborhood conditions need to be provided. The authors cite a previous literature (Van Holle et al, 2014), but that reference has limited information about the neighborhood type. For example, four neighborhood types considering walkability and income (HH, HL, LH, HH), needs to be explained in depth along with each type’s physical environmental factors.
- This might be beyond the scope of this paper, but if it is possible, the authors might consider adding another analysis process, in which neighborhood types’ differences and significances in the relationships between determinants and moderators concerning environment-AT associations.
- Discussion part is interesting. Among others, contradictory result (Line 319) needs further explanation. It might be caused from 'self-reporting' method, in which people confuse or ignore walking and other activities related to public transport. Perceived physical activity and objectively active transport could to be explored in the future studies.
Reviewer 3 Report
I have reviewed the manuscript entitled “Psychosocial and social environmental factors as moderators in the relation between the objective environment and older adults’ active transport.” The article is very interesting, logically organized and well written. The method is clearly described and logically grounded in literature’s review. The paper is publishable, if the following comments are properly addressed in the revised version.
One is statistical significance of coefficients, i.e. whether 503 house location data points are sufficient to fit multiple models containing too many model parameters, and meaningfully interpret the coefficients. Another one is whether twenty neighborhoods does have enough variations to disentangle the effects of environmental factors. This may well be ok, but insufficient information is provided to assess this.
Several factors known to affect physical activity are missing from this study. The dependent variable, AT, (walking and cycling for transport) is affected by vehicle traffic and slope. While land use mix entropy is included in the model, other factors relating to pleasantness of routes for walking and cycling such as slope, retail frontage, street lighting, and safety are absent. Absence of factors known to be important shouldn’t affect methodological conclusions, but thought should be given as to whether we can draw policy conclusions from the model without including these factors in some form. The authors need to at least acknowledge the missing factors and discuss how their absence might bias the findings.
Another issue is the transferability of results; that is, whether the findings are applicable beyond Belgium? In my opinion the findings are publishable whether or not this is the case, but the authors should make clear which of their findings they expect to be specific to Belgium versus which they expect to be transferable to other countries or cities.
Round 2
Reviewer 2 Report
file attached

Author Response
The sampling or the selection of the test subjects, was done on the basis of four different neighbourhoods: high income/high walkability, high income/low walkability, low income/high walkability, and low income/low walkability. This was done to ensure that there were roughly equal numbers of participants recruited from each neighbourhood. Thereafter, all these participants were processed together in one data file. Consequently, for the further analyses, the type of neighbourhood (i.e. regarding the four different types of neighbourhoods: HH, HL, LH, HH) in which one lives at neighbourhood level was not taken into account.
For our study, the individual GIS data (i.e. at individual level) was examined per subject. So the stance of this paper, revealed in the abstract and discussions, carries indeed certain neighborhood-type approaches, but at individual level. That the individual GIS data is used, is also exactly what is innovative about this study in comparison to other studies.